# Variations in brain defects result from cellular mosaicism in the activation of heat shock signalling

Seiji Ishii[1,*], Masaaki Torii[1,2,3,*], Alexander I. Son[1], Meenu Rajendraprasad[1,4], Yury M. Morozov[2], Yuka Imamura Kawasawa[5,6,7], Anna C. Salzberg[7], Mitsuaki Fujimoto[8], Kristen Brennand[9,10], Akira Nakai[8], Valerie Mezger[11,12,13], Fred H. Gage[10], Pasko Rakic[2] & Kazue Hashimoto-Torii[1,2,3]

Repetitive prenatal exposure to identical or similar doses of harmful agents results in highly variable and unpredictable negative effects on fetal brain development ranging in severity from high to little or none. However, the molecular and cellular basis of this variability is not well understood. This study reports that exposure of mouse and human embryonic brain tissues to equal doses of harmful chemicals, such as ethanol, activates the primary stress response transcription factor heat shock factor 1 (Hsf1) in a highly variable and stochastic manner. While Hsf1 is essential for protecting the embryonic brain from environmental stress, excessive activation impairs critical developmental events such as neuronal migration. Our results suggest that mosaic activation of Hsf1 within the embryonic brain in response to prenatal environmental stress exposure may contribute to the resulting generation of phenotypic variations observed in complex congenital brain disorders.

[1] Center for Neuroscience Research, Children's National Medical Center, Washington, District of Columbia 20010, USA. [2] Department of Neurobiology and Kavli Institute for Neuroscience, Yale University School of Medicine, New Haven, Connecticut 06510, USA. [3] Department of Pediatrics, Pharmacology and Physiology, School of Medicine and Health Sciences, George Washington University, Washington, District of Columbia 20052, USA. [4] Department of Biomedical Engineering, School of Engineering and Applied Science, George Washington University, Washington, District of Columbia 20052, USA. [5] Department of Pharmacology, Pennsylvania State University College of Medicine, 500 University Dr., Hershey, Pennsylvania 17033, USA. [6] Department of Biochemistry and Molecular Biology, Pennsylvania State University College of Medicine, 500 University Dr., Hershey, Pennsylvania 17033, USA. [7] Institute for Personalized Medicine, Pennsylvania State University College of Medicine, 500 University Dr., Hershey, Pennsylvania 17033, USA. [8] Department of Biochemistry and Molecular Biology, Yamaguchi University School of Medicine, Ube 755-8505, Japan. [9] Department of Psychiatry and Neuroscience, Icahn School of Medicine at Mount Sinai, New York 10029, USA. [10] Salk Institute for Biological Studies, Laboratory of Genetics, La Jolla, California 92037, USA. [11] CNRS, UMR7216 Epigenetics and Cell Fate, Paris 75205, France. [12] University Paris Diderot, 75205 Paris, France. [13] Département Hospitalo-Universitaire DHU PROTECT, Paris 75019, France. * These authors equally contributed to this work. Correspondence and requests for materials should be addressed to P.R. (email: pasko.rakic@yale.edu) or to K.H.-T. (email: khtorii@cnmc.org).

Environmental stressors are critical factors impacting the proper development of the nervous system. The cerebral cortex, the intricate outer layer of the cerebrum whose formation relies on a complex series of coordinated developmental events[1,2], is particularly vulnerable to disturbances in the prenatal environment[3–5]. Clinical and epidemiological studies have identified a variety of such stressors including heavy metals and alcohol to enhance the risks of neuropsychiatric diseases[6].

Exposure to environmental stressors results in immediate alterations in the molecular landscape of afflicted cells[7]. The reduced expression of many genes affect brain development deleteriously[8,9], while the upregulation of others are involved in pathological processes such as cell death[10,11]. To mitigate these outcomes, cells launch protective signalling events simultaneously that attempt to counter these effects. One such mechanism is the Hsf1-Heat shock protein (Hsp) signalling pathway, a well-known stress-induced chaperone pathway that serves as a universally conserved cytoprotective mechanism against multiple forms of cellular stress across several organisms ranging from bacteria to humans[12–14]. Hsf1 is activated by heat stress or other stimuli through sequential protein modification events including phosphorylation, sumoylation, trimerization, and nuclear translocation. Activation then triggers the transcription of Hsps and other downstream target genes.

Our recent study has revealed that Hsf1–Hsp signalling is commonly activated on prenatal exposure to various distinct types of environmental stressors including ethanol (EtOH), methyl mercury (MeHg), and maternal seizures, and is required for reducing the risk of cortical malformations such as leptomeningeal heterotopias, thereby reducing susceptibilities to epilepsy[15]. In addition, the intercellular variability in transcriptional stress responses due to environmental stressors, including that of Hsf1–Hsp signalling, is augmented in pathological conditions such as aging and neuropsychiatric diseases in higher-order organisms[15,16]. Interestingly, unicellular microbial populations rely on this non-genetic cell-to-cell gene expression variability for the survival of a clonal population in response to abrupt environmental changes[17–19]. However, the roles of mosaic cellular responses in brain pathogenesis in vertebrates are still unclear.

This study identifies the probabilistic activation of Hsf1–Hsp signalling on exposure to environmental stressors and the distinct phenotypic consequences resulting from differential activation levels in cortical development. These results suggest that heterogeneous events of abnormal development may occur probabilistically, accounting for individually distinct patterns of focal cortical malformations in the cortex exposed to similar levels of environmental challenges.

## Results

**Probabilistic HSF1 activation in human neural progenitors**. To test whether the HSF1 activation levels of human neural progenitor cells (hNPCs) in response to environmental stress are determined in a random manner or correlated with their geographical location in the dish, we have taken advantage of the single molecule fluorescence *in situ* hybridization (smFISH) method for visually analysing mRNA expression levels of *HSP70*, a major downstream transcriptional target of HSF1 in each single cell (Fig. 1). Given that *HSP70* mRNA is very stable (with a half-life of more than 8 h) under stress exposure[20], smFISH analysis after a 3 h exposure provides an overall picture of the average transcriptional activity of HSF1 during the stress exposure in each hNPC. For this analysis, we used three types of environmental stress: EtOH, hydrogen peroxide ($H_2O_2$) and MeHg, all of which induce oxidative stress at certain

concentrations. The highest concentration for each agent was set to a subthreshold level, which is half of the concentration that induces massive (more than 20% of the total number of cells) cell death.

After 3 h of stress exposure, we observed that the number of *HSP70* mRNA particles in each cell varied among the hNPCs regardless of the type of environmental stressors (PBS control did not induce *HSP70* expression; Fig. 1a, Supplementary Fig.1). This effect was dosage dependent, as the variability was higher when exposed to environmental stressors at increased concentrations (Fig. 1c,d). The levels of activation appeared to be random as they differed across hNPCs in close proximity regardless of their location within the culture dish (peripheral, central and so on), hence sharing the same microenvironment. The cell-to-cell variability was similarly observed for hNPC lines derived from differently induced pluripotent stem cell (iPSC) lines (Fig. 1c,d). The variability was maintained during a long period of exposure (Supplementary Fig. 2a–c), supporting the notion that this effect may have significant consequences on brain development[15]. No significant effects were observed in the variability of the expression of the housekeeping gene *GAPDH* by environmental stressors (Fig. 1b-d).

We next confirmed that the probability of HSF1 activation was independent of the cell's location in the dish through statistical simulations. For each dish, the spatial patterning of *HSP70* expression was shuffled to generate a set of randomized patterns. The Pearson correlation coefficient was calculated for every pair within the data set after random permutation of the experimental data, and the average of the correlation coefficients derived from a thousand of the permutations was plotted. The resulting data for each environmental stressor shows a normal distribution, and the average coefficient of the simulated data set shows no significant difference compared with that of the experimental data (Fig. 1e–g). Consistent with previous studies[21], the correlation between the *HSP70* mRNA expression level and the cell cycling status was not evident (Supplementary Fig. 2d). These results suggest that the Hsf1–Hsp70 signalling is mainly activated at different magnitudes in a probabilistic fashion among hNPCs *in vitro*.

**Heterogeneous Hsf1 activation in mouse embryonic cortex**. We next examined the Hsf1 activation pattern *in vivo* by analysing the developing mouse cerebral cortex. For this analysis, we made a reporter plasmid that includes a red fluorescent protein (RFP:DsRed2) encoding gene under the control of the *Hsp70* promoter (pHSE–RFP [HSE: Heat Shock Element], Supplementary Fig. 3a,b). This *Hsp70* promoter contains two Hsf1 binding sites, driving RFP (DsRed2) expression upon Hsf1 activation. Initially, we used *in utero* electroporation (IUE)-mediated transfer of the reporter constructs into cortical neural precursor cells (NPCs) generating excitatory neurons. pHSE–RFP was co-electroporated at embryonic day (E) 14 into the cortex with pCAG–GFP, which labels all the electroporated cells with green fluorescent protein (GFP) expression. This procedure was followed by daily intraperitoneal injections of EtOH as a representative stressor (or control PBS) which were started 6 h post-electroporation. The HSE–RFP reporter expression was rarely detected in the embryos from PBS-injected dams 1 day after the last injection at E16 ($<0.01\%$ of total electroporated cells express RFP), whereas RFP was expressed in most of the embryos from EtOH-injected dams (Fig. 2b,c). In embryos from EtOH-exposed dams, $23.6 \pm 1.6\%$ (s.e.m.) of the total electroporated ($GFP^+$) NPCs and descendent neurons expressed the HSE–RFP reporter in sporadic patterns, showing no indications for specificity in cortical regions (Fig. 2a–h,

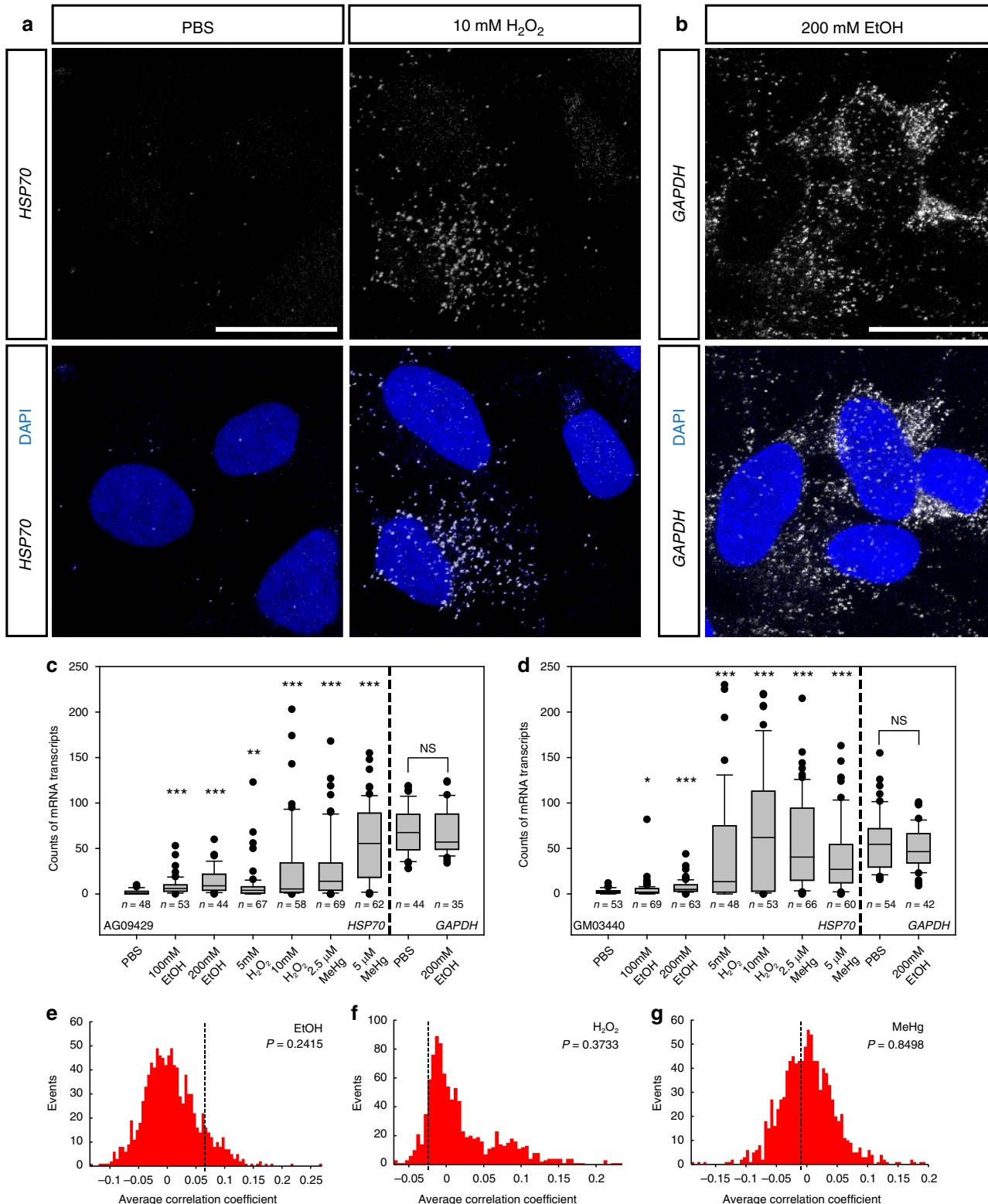

**Figure 1 | Variable levels of activation of Hsf1–Hsp signalling by environmental stress.** (**a**,**b**) Representative smFISH images of *HSP70* mRNA (**a**) and *GAPDH* mRNA (**b**) in an hNPC line (AG09429). (**c**,**d**) Quantification of *HSP70* and *GAPDH* mRNA puncta in hNPCs (AG09429 and GM03440) exposed to the indicated reagents. In the box plot, the line within the box indicates the median; the upper and lower edges of the box represent the 25th and 75th percentiles, respectively. Error bars indicate the 10th and 90th percentiles and each dot indicates cells within the top/bottom 10th percentiles. The variability of *HSP70* mRNA particles was significantly increased in response to all these reagents compared to that with PBS exposure (*$P < 0.05$, **$P < 0.01$, ***$P < 0.001$ by Levene's test. More than 35 cells in different positions within the dishes were measured in each group). (**e–g**) The correlation coefficients of cellular positions in the dishes and *Hsp70* mRNA levels are compared between the experimental data (broken lines) and simulated data sets representing probabilistic events (red histograms). The presented *P* values of one-sample *Z*-test indicate that the levels of the Hsf1–Hsp 70 signalling activation in each cell are randomly determined regardless of the positions in the culture dish. Scale bars, 0.02 mm.

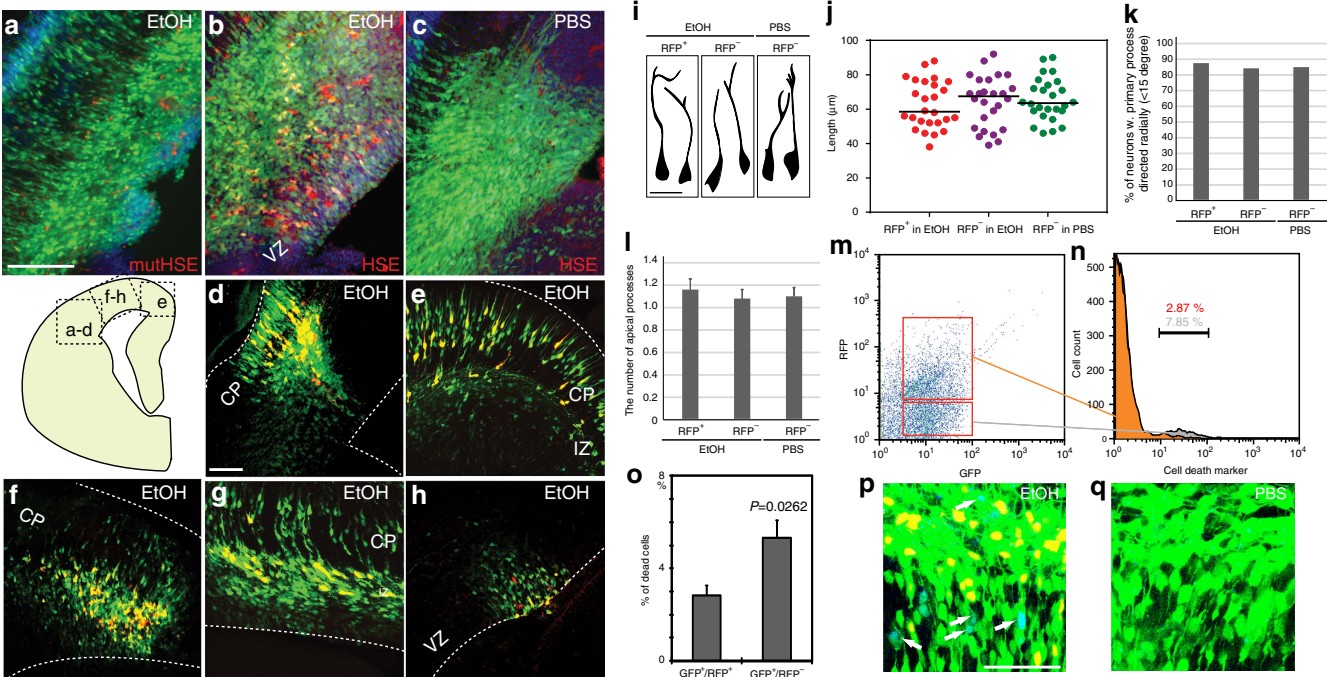

**Figure 2 | Normal development of HSE–RFP reporter⁺ neurons and increased cell death of reporter⁻ cells in IUE-based reporter analysis.**
(**a–c**) pHSE- (**b,c**) or pmutHSE- (**a**) RFP was electroporated with pCAG–GFP into the mouse (C57NL/6J) lateral cortex at E14, followed by daily EtOH (**a,b**) or PBS (control, c) administration to the dam until E16. The reporter expression (red) was observed in NPCs and neurons by electroporation of p HSE–RFP followed by EtOH administration (**b**). Electroporation of pmut HSE–RFP plasmid that contains point mutations in HSF1 binding sites in the promoter (Supplementary Fig. 3) resulted in reporter expression in less than 0.1% of total electroporated (GFP⁺) cells in the cortex exposed to EtOH (n = 22). Similarly, electroporation of p HSE–RFP in *Hsf1* KO mice resulted in no reporter induction by EtOH exposure (n = 3/3). (**d–h**) The HSE–RFP reporter expression was observed in NPCs and neurons throughout the cortex, including lateral (**d**), medial (**e**) and dorsal (**f–h**) regions. CP: cortical plate, IZ; intermediate zone, VZ: ventricular zone. Scale bar, 250 μm. (**i**) Normal morphology of HSE–RFP ⁺ neurons in IUE approach. Representative morphologies of RFP⁺/GFP⁺ and RFP⁻/GFP⁺ neurons in the CP exposed to EtOH or PBS (control) as indicated. (**j**) The length of the primary process was plotted for neurons in the indicated groups, showing no significant differences between the groups. (**k**) The percentage of neurons with a radially-directed (<15 degrees) primary process. (**l**) The number of apical processes was quantified for neurons in the indicated groups. No significant differences were found between the groups. (**m–o**) Flow cytometric analysis of the electroporated cells (**m**), revealed that HSE–RFP ⁺ cells (orange in n) contain less dead cells than HSE–RFP ⁻ cells (gray in n) in the cortex exposed to EtOH. The mice were electroporated with p HSE–RFP and pCAG–GFP at E14 followed by daily EtOH administration until E16. Cells were then dissociated and stained with a cell death marker (7-AAD). (**o**) Percentages of the cells positive for cell death marker in HSE–RFP -positive and –negative cells. P<0.01 by t-test (n = 7 embryos). (**p,q**) Consistent with the data quantified with 7-AAD (**m–o**), activated Caspase-3 (blue) is observed frequently in HSE–RFP ⁻/GFP⁺ electroporated cells (**p**, shown in white arrows), but not in HSE–RFP ⁺/GFP⁺ in EtOH (**p**) or HSE–RFP ⁻/GFP⁺ in PBS-exposed (**q**) cortices. Scale bar, 50 μm.

Supplementary Table 1). No induction of the HSE–RFP reporter in *Hsf1* knockout (KO) mutant embryos exposed to EtOH was observed (n = 3/3 embryos), confirming that reporter induction was caused solely by Hsf1 activation. The morphology and positioning of the reporter-positive cells appeared to be normal (Fig. 2i–l). However, flow cytometric analysis revealed that the HSE–RFP reporter-positive cell population contained significantly fewer dying cells compared with the reporter-negative cell population (Fig. 2m–q), consistent with the protective roles of activated Hsf1 (ref. 15).

As an alternative approach, we generated HSE–RFP reporter transgenic mouse lines harboring the same reporter construct and obtained three founder lines. Using the same alcohol exposure regimen as we used for IUE experiments, we compared reporter sensitivities in these mice to that of electroporated mice. HSE–RFP reporter-expressing cells were FACS-sorted from EtOH-exposed cortices at E14 and E16, and the Hsf1 activity was determined based on the *Hsp70* mRNA expression levels by real-time quantitative reverse transcription–PCR (qRT–PCR). The data revealed that the cells detected by the reporter in transgenic founder lines had higher levels of *Hsp70* expression on average than those detected in the electroporated mice (Fig. 3a).

No significant differences were detected between the founder lines[22]. Given that the electroporation-based reporter assay shows higher sensitivity in general[23], this indicates that the transgenic reporter detects only cells in which Hsf1–Hsp signalling is activated at high levels.

In contrast to the electroporation-based reporter analysis, reporter expression is observed in only a subset of embryos in each litter of the transgenic reporter mice. As shown in the representative images, the reporter expression is also restricted in limited cortical regions, the pattern of which varies among embryos with no apparent regional preferences (Fig. 3b–d). The reporter expression is not correlated to the position of the embryo in the uterus or in the location of the EtOH injection within the dam. Gavage administration of EtOH induced similar heterogeneous reporter expression patterns. Since RFP (DsRed2) protein is stable and detectable for more than 4 days following induction (for example, Fig. 3g), the heterogeneous reporter expression is not due to capturing oscillatory activities of HSF1 transcription. FACS analysis revealed that the reporter-expressing cells consist of both Tuj1⁺ immature neurons and Nestin⁺ NPCs, but few NeuN⁺ mature neurons (Fig. 3f).

As the genomic insertion site of the transgene may compromise optimal observation of heterogeneous gene expression, we defined the genomic loci of transgene integration by paired-end whole genome sequencing of HSE–RFP transgenic mice. Of the two insertion sites identified, one is at the intergenic region on Chromosome 10, where a read pair is aligned to HSE and an intrinsic genomic region (Supplementary Fig. 4a). The second is on Chromosome 5, where two read pairs are aligned to either HSE or the DsRed2 sequence on one side, and to intrinsic sequences within a 3 kb genomic region on the other side (Supplementary Fig. 4b).

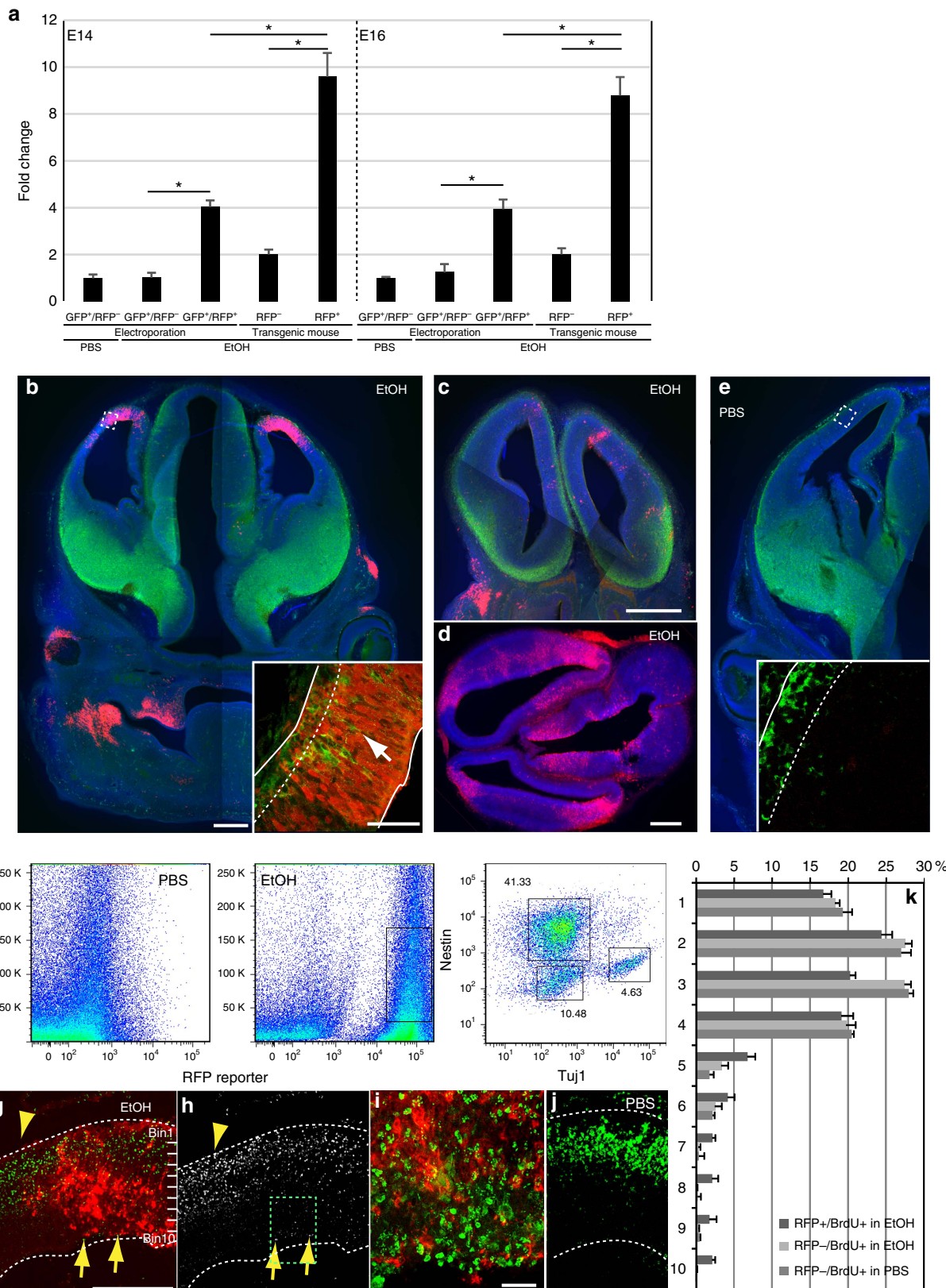

The sequences of these read pairs revealed that the transgene is inserted within an intron of the *Tctn2* gene, but in the opposite direction of its transcription. These results indicate that the observed heterogeneous transgene expression is independent of the influence of its transgene locus.

Heterogeneous activation of Hsf1–Hsp signalling in cortical cells was also confirmed by sporadic distribution of cells labelled with another molecular indicator, nuclear translocation of the HSF1, in both mouse and human embryonic cortices exposed to EtOH (Supplementary Fig. 5). Similar heterogeneous reporter expression patterns across embryos, as well as among cells within the brain and other tissues in each embryo, have been observed by prenatal exposure to other environmental factors including sodium arsenite, Pentylenetetrazol (PTZ)-induced maternal seizure, and lipopolysaccharide (LPS)-induced maternal inflammation[22].

Altogether, these results demonstrate cellular mosaicism after Hsf1–Hsp signal activation in the embryonic cerebral cortex. Consistent with the results obtained using *Hsf1* knockout mice exposed to environmental stressors[15], our results also suggest that cells with low levels of activation face a higher risk of cell death under exposure to EtOH.

**Excess Hsf1 activation impairs migration of cortical neurons.** We observed the regions containing HSE–RFP reporter-expressing cells was often associated with delayed migration of immature neurons (inset in Fig. 3b). To formally test this possibility, Bromodeoxyuridine (BrdU) pulse labelling (a BrdU injection 30 min before the second injection of EtOH at E15 and observation at E18) was performed. The disturbed cell migration was observed in the domains with a significant number of reporter[+] cells (Fig. 3g–j). Similar migration defects of cortical neurons associated with the HSE–RFP expression have been observed by prenatal exposure to the drug suramin[22]. This suggests that the cells with high levels of Hsf1–Hsp signalling activation induced by EtOH and other environmental stressors undergo delayed migration. This appears independent of cell death, which does not differ between HSE–RFP reporter[+] and reporter[-] cells from the reporter transgenic embryos (Supplementary Fig. 6).

To examine whether the high-level activation of Hsf1–Hsp signalling causally affects the migration of cortical neurons, an expression construct of the constitutively active form of HSF1 (caHSF1) or pCAGIG control plasmid[24] was introduced into the cortex of wild-type mice by IUE at E14, and GFP[+] cells were dissociated 24 h post-electroporation. Quantification of *Hsp70* mRNA particles in individual cells using smFISH revealed that the activation level of Hsf1–Hsp signalling by caHSF1 was

comparable to the top 5% levels induced by EtOH exposure (Fig. 4a,b, compared with Fig. 1). Thus, the expression of caHSF1 recapitulated the physiologically relevant level of HSF1 activation that occurred in a small population of cells in the events of challenged exposure. In the control electroporation, GFP[+] electroporated cells were located around layers II-IV at postnatal day (P) 14 (Fig. 5a). In all cases of caHSF1 introduction, however, GFP[+] cells formed clusters beneath the gray matter (Fig. 5b). The full-length wild-type and the loss-of-function type R71A mutant, both of which lack the potency to sufficiently activate Hsf1–Hsp signalling, did not exhibit significant effects (Fig. 5c,d). EtOH exposure following caHSF1 electroporation did not have any obvious additional effects on heterotopia formation (Fig. 5c,d). BrdU pulse labelling at E15, 1 day after IUE, and immunostaining for molecular markers such as NeuN (for mature neurons) and Cutl1 (upper layer neurons) at P14 indicated that caHSF1 expression did not affect neuronal differentiation/maturation and subtype specification (Fig. 6a–l). The heterotopia began to develop as early as embryonic stages (Figs 5g, 6m,n). Tuj1 labelling at E18 and Terminal deoxynucleotidyl transferase dUTP nick end labelling (TUNEL) at E16 showed that neither neuronal differentiation nor cell death appeared to be affected in the electroporated cells (Fig. 5f,g, and Supplementary Fig. 7a). Disruptions of the ventricular surface, similar to those observed by *in utero* exposure to EtOH (Figs 3b,g, 2a, Supplementary Fig. 5h), were frequently found at the caHSF1-electroporated region (Fig. 5e,h, Supplementary Fig. 7a). Furthermore, electron microscopy analysis one day after electroporation of caHSF1 revealed that an impairment of adherens junctions in the GFP[+] NPCs aligned on the ventricular surface was involved in these disruptions[25,26] (Fig. 5i). Immunohistochemistry for Nestin also showed abnormalities in the radial morphology of caHSF1-expressing NPCs (Fig. 5h). These results suggest that the disrupted radial glial scaffold and the ventricular lining due to excess activation of HSF1 in NPCs might contribute to the impaired neuronal migration non-cell autonomously, as implied by the migration defects of non-electroporated cells (Fig. 6b–h) or reporter-negative cells in the EtOH-exposed cortex (Fig. 3i).

To formally define the relative contribution of the cell-autonomous effect of excess activation of HSF1 to the impaired neuronal migration, we expressed caHSF1 exclusively in migrating immature neurons using the CALSL/Tα1-Cre system[27–29]. The results showed that excess HSF1 activation in migrating immature neurons was sufficient to impair the radial migration in a cell-autonomous manner (Fig. 5j). Interestingly, overexpression of *Hsp70*, a canonical downstream target of HSF1, did not disrupt the migration (Supplementary Fig. 7b),

**Figure 3 | Heterogeneity in the reporter expression of Hsf1–Hsp signalling and the migration deficiency in reporter[+] cells in the cortex.** (**a**) Different sensitivity for detection of Hsf1–Hsp signalling activation between IUE- and transgenic-based reporter assays. qRT-PCR of *Hsp70* was performed with the total RNA extracted from reporter-positive and -negative cells sorted by a flow cytometer 8 h after administration at E14 (left) or E16 (right). Data are represented as mean ± s.e.m. *P < 0.0001 by *t*-test from three independent experiments using multiple embryos from several dams. (**b,c,d**) Heterogeneous reporter expressions of Hsf1–Hsp signalling driven by EtOH exposure at E15 (**b**) or 16 (**c,d**) in three HSE–RFP reporter transgenic mice 1 day after last exposure. Each sample shows a distinct reporter expression pattern. Reporter-expressing regions contain Tuj1[+] immature neurons (green) incorrectly positioned in the ventricular zone (arrows in the inset in **b**). The square in **b** indicates the region from which the inset was taken. The dotted line in the inset in b shows the boundary of the cortical plate and ventricular zone. Approximately 9% of the total analysed embryos showed robust reporter expression (n = 10/115 embryos) (green: Tuj1, red: HSE–RFP, blue: DAPI). (**e**) No reporter expression after PBS (control) exposure. (0/137 embryos). (**f**) Flow cytometric analysis of reporter expressing cells dissociated from E16 cortices 1 day after the last exposure. The percentages of Nestin[+]/Tuj1[-], Nestin[-]/Tuj1[+] and Nestin[-]/Tuj1[-] cells were determined after gating of the RFP[+] cells as indicated with a square in the EtOH-exposed sample. The objects showing larger FSC profiles are cellular debris. (**g–j**) Birthdate labelling with BrdU (green in (**g,i,j**), white in (**h**)) at E15 shows delayed migration of BrdU[+] cells at E18 in the region containing HSE–RFP[+] cells (arrows) compared with the surrounding region (arrowheads), after daily administration of EtOH at E14-16. **i** is a higher magnification view of square in **h**. (**k**) Quantification was performed by identifying the domains that contain RFP[+] cells throughout radial axis in somatosensory cortical regions. The RFP[-] cells were counted in the adjacent regions in EtOH exposed samples and the corresponding regions in PBS-exposed samples (RFP[-]/BrdU[+] in PBS). *P < 0.001, repeated measures ANOVA, n = 3 each. Scale bars, 0.5 mm (**b–h,j**), and 50 μm (in the inset in **b,e,i**).

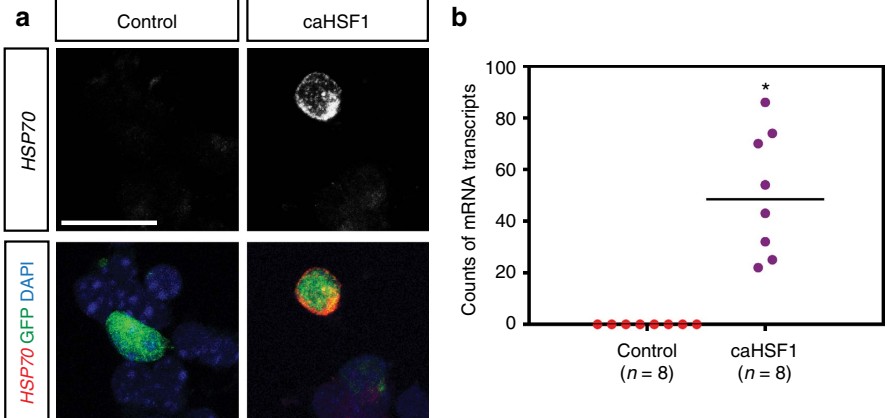

**Figure 4 | IUE-mediated expression of caHSF1 recapitulates the physiologically relevant levels of HSF1 activation elicited by environmental challenges.** (**a**) Representative images of *Hsp70* mRNA expression in pCAGIG (control) or pCAGIG-caHSF1 electroporated cells. The plasmids were introduced into mouse frontal cortices by IUE at E14. The electroporated cells were dissociated 24 h post-electroporation, and plated on culture dishes. 3 h later, *Hsp70* mRNAs were detected by smFISH. (**b**) The graph shows the number of *Hsp70* mRNA particles in the control- and caHSF1-electroporated cells. caHSF1 overexpression results in significantly higher numbers of *Hsp70* mRNAs compared with the control (*$P < 0.001$ by Mann-Whitney U-Test). Importantly, these numbers fall in the range of the top 5% and 20% of those induced by EtOH and MeHg, respectively (Fig. 1 and Supplementary Fig. 1). Scale bar, 0.02 mm.

suggesting that the impaired neuronal migration was mediated by non-canonical downstream targets of Hsf1.

**Excess Hsf1 activation induces permanent cell dispositioning.** To better mimic the temporal HSF1 activation by environmental challenge exposures in the cases of human patients, we limited the expression of caHSF1 in a specific time window using the 4-OHT (active metabolite of tamoxifen)-inducible Cre/LoxP system. E14 cortices were electroporated with plasmids containing the constructs CAG-promoter-loxP-FLAGcaHSF1-loxp-GFP (CALHLG), pCAG-ER<sup>T2</sup>-Cre-ER<sup>T2</sup> (CAG-CreERT2), and pCAG-RFP (Fig. 7a). Expressing caHSF1 until after birth (to P0 or P3) caused the formation of large heterotopias compared with the vehicle (Fig. 7e–i). Although very low levels of caHSF1 expression with a 4-OHT administration immediately after after caHSF1 introduction (6 h after IUE at E14) did not form heterotopias (Fig. 7b,h,i), expression of caHSF1 until E15 or E16 was sufficient to lead to incomplete neuronal migration (Fig. 7c,d,h,i). No migration defects by electroporation of pCALNL-GFP with CAG-CreERT2 and pCAG-RFP followed by 4-OHT injection were observed (Fig. 7j). These results show that even a short period of HSF1 overactivation during prenatal development causes critical neuronal migration deficiency and confirms that the severity of deficiency critically depends on the duration of HSF1 overactivation. This is in line with the risk of the cortical maldevelopment which is proportional to levels/duration of exposure in general.

**Reduction of Hsf1 reverses impaired cortical cell migration.** To assess whether Hsf1 overactivation is not only sufficient but also required for the migration deficits, we next tested whether the reduction of Hsf1 activity could mitigate the migration defects elicited by EtOH. Since EtOH-induced Hsf1 overactivation and migration deficits occur sporadically in the cortex (Fig. 3), it is impossible to predict such domains to reduce Hsf1 activity *in vivo*. We therefore performed an *in vitro* migration assay with cortical NPCs dissociated from E13 mouse embryos (Fig. 8a). To examine the cell autonomous effects of *Hsf1* knockdown, NPCs electroporated with *Hsf1* short hairpin RNA (shRNA)[15] (or control shRNA) and non-electroporated NPCs

were cultured together to generate neurospheres. Once the diameter of the neurospheres reached 50–100 μm, the cells were placed on coated plates to migrate out from the neurosphere (Supplementary Fig. 8a) for 3 or 6 h in the condition with either EtOH or PBS. The number of Nestin<sup>+</sup> NPCs did not differ between those exposed to PBS or EtOH (Supplementary Fig. 8b,c). However, the EtOH exposed group displayed a failure of migration out of the neurosphere in a significant number of NPCs electroporated with the control shRNA, which was not observed in the PBS controls (Fig. 8c–f). smFISH analysis showed that the level of Hsf1 activation assessed by the number of *Hsp70* mRNA particles in each NPC and the distance of migration from the center of the neurosphere were negatively correlated (Fig. 8b). The rate of cell death and cell cycling did not differ between NPCs that migrated out of neurospheres and those that remained within neurospheres (Supplementary Fig. 8d,e). Introduction of *Hsf1* shRNA caused a sharp contrast, as the percentage of the NPCs that had migrated longer distances was increased significantly in cultures with EtOH (Fig. 8c–f). *Hsf1* shRNA-mediated amelioration of migration defects was lost by overexpression of human HSF1 that is not targeted by the mouse *Hsf1* shRNA, confirming the specificity of the effect of *Hsf1* shRNA (Supplementary Fig. 9). These results indicate that excess activation of Hsf1 is directly responsible for EtOH-induced migration deficits.

**Discussion**
This present study demonstrates that the level of activation of Hsf1–Hsp signalling on prenatal exposure to environmental stress is highly variable among cortical NPCs and immature neurons. We recently demonstrated that insufficient Hsf1–Hsp signalling activation on exposure to environmental challenges causes a higher incidence of leptomeningeal heterotopia, as well as defects in neuronal production and survival in the cerebral cortex, all of which are associated with increased risk of seizures[15]. This present study shows that excessive activation of Hsf1–Hsp signalling also leads to different abnormalities in cortical development such as neuronal migration defects and periventricular heterotopia. These findings suggest that the heterogeneous focal cortical malformations caused by discordant activation of the Hsf1–Hsp

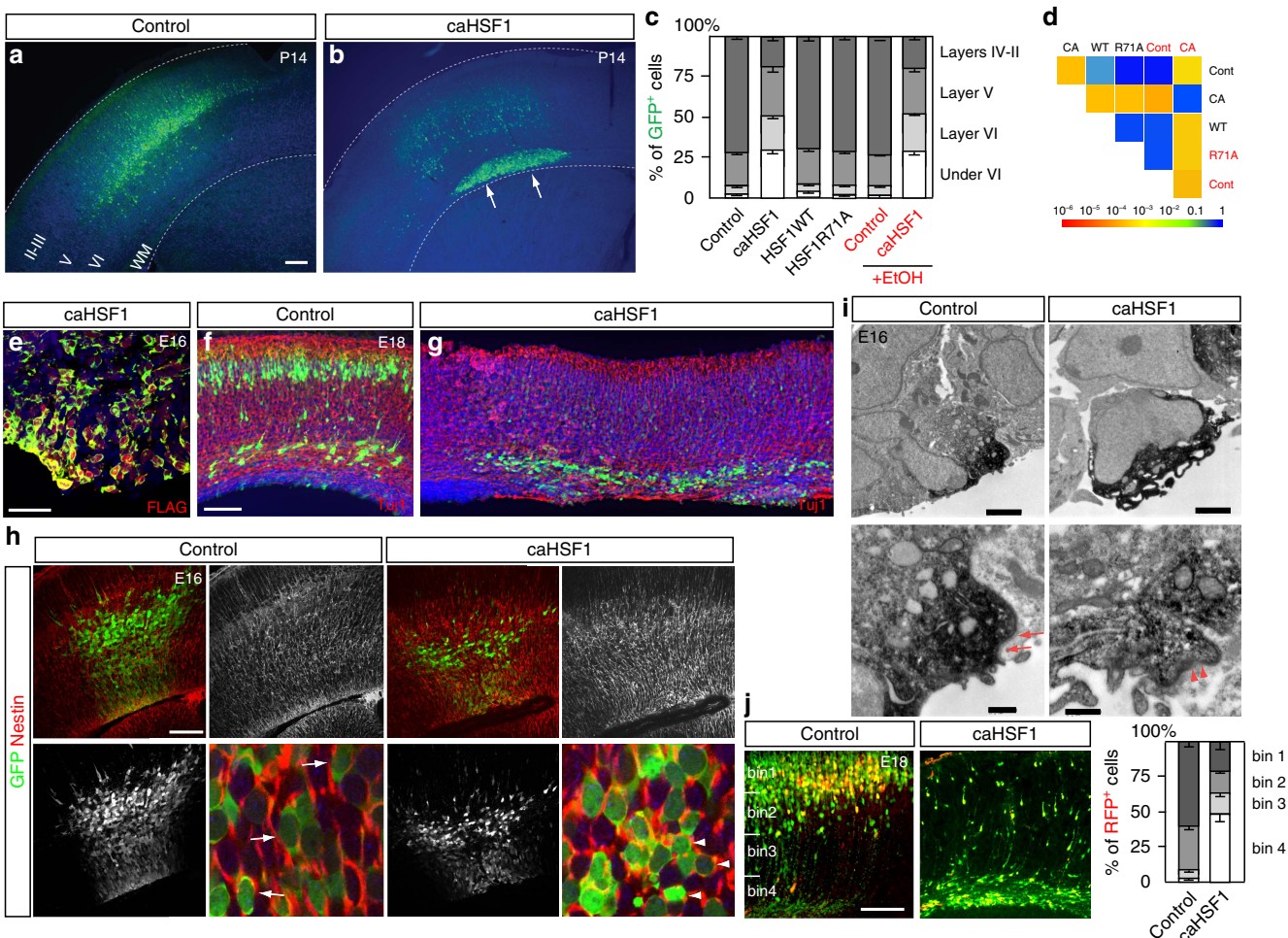

**Figure 5 | Excess activation of Hsf1 causes heterotopia formation.** (**a,b**) The P14 cortex electroporated with the control (**a**) or caHSF1 expression (**b**) plasmid at E14. caHSF1 expression causes the formation of periventricular heterotopia. Blue: nuclei counter staining. (**c**) Distribution of GFP$^+$ electroporated cells transfected with indicated constructs. Data are represented as mean ± s.e.m. F(3,48) = 1197, P < 0.0001 by repeated measures ANOVA among groups without EtOH exposure. P < 0.001 by *post hoc* Tukey test between caHSF1 and other HSF1 forms or control. F(3,24) = 80.59, P < 0.0001 between caHSF1 and control comparison under EtOH exposure. (**d**) Pairwise comparisons of cumulative radial distributions using Kolmogorov–Smirnov (K–S) test. Each P-value is shown in a color according to the scale at the bottom. (**e**) Confirmation of caHSF1 (with a FLAG-tag) expression in the GFP$^+$ electroporated cells by FLAG immunohistochemistry. Blue: nuclei counter staining. (**f–h**) Tuj1 (**f,g**) and Nestin (**h**) immunohistochemistry at indicated ages following IUE at E14. Arrows and arrowheads in **h** indicate oval-shaped, radially oriented NPCs in the control, and abnormal round-shaped, non-radially oriented caHSF1-expressing NPCs, respectively in the VZ. (**i**) Immuno-electron microscopy of control- and caHSF1-electroporated NPCs at the ventricular surface. Electroporated cells were visualized by GFP immunohistochemistry using DAB (black). Complete loss of adherens junctions is accompanied by the detachment of caHSF1-expressing NPCs from surrounding non-electroporated cells. The remaining disorganized adherens junctions are revealed by decreased electron density (arrowheads) compared with normal adherens junctions in the control (arrows). (**j**) Using the CALSL/Tα1-Cre system, exogenous genes (RFP only (control) or RFP plus caHSF1) were expressed exclusively in immature neurons. The GFP plasmid (pCAGIG) also was introduced to label all electroporated cells (n = 6 embryos). caHSF1 expression interferes radial migration of neurons cell-autonomously. Radial distribution of RFP$^+$ neurons (graph) shows significant difference between control and caHSF1 electroporation (F(3,24) = 244.3, P < 0.0001 by repeated measures ANOVA. P < 0.0001 by K-S test). Data are represented as mean ± s.e.m. Scale bars, 0.3 (**a,b**), 0.04 (**e**) and 0.2 (**f–h,j**) mm, and 5 (**i**, upper panels) and 0.2 (**i**, lower panels) μm.

signalling in response to prenatal environmental stress may be an underlying mechanism in human clinical cases associated with phenotypic variations in which various types of focal cortical malformations exist in different cortical regions in each brain[30,31] (Fig. 8g). Similar to *de novo* somatic mutations during the early stage of brain development[32,33], small epigenetic mosaicism during the early developmental stage is likely to significantly impact later brain development, resulting in mosaic disorganization reported in the brains with psychiatric disorders[34].

Consistent with previous studies demonstrated in microbes[35,36], the present study suggests probabilistic activation of Hsf1–Hsp

signalling in the NPCs of the mammalian brain. The results of the whole genome sequencing of the reporter transgenic mice (Supplementary Fig. 4) suggest reporter expression to occur ubiquitously throughout the body. Indeed, we observed broad expression in many tissues/organs including the eye, heart, liver and skin[22]. However, the expression patterns across various tissues and organs vary between animals. Potential factors for these differences include differential circulation/accumulation of EtOH between tissues/organs, differential activation of Hsf1–Hsp signalling between tissue/organ-specific cell types, and stochastic response of cells across tissue/organs. The results of our *in vitro*

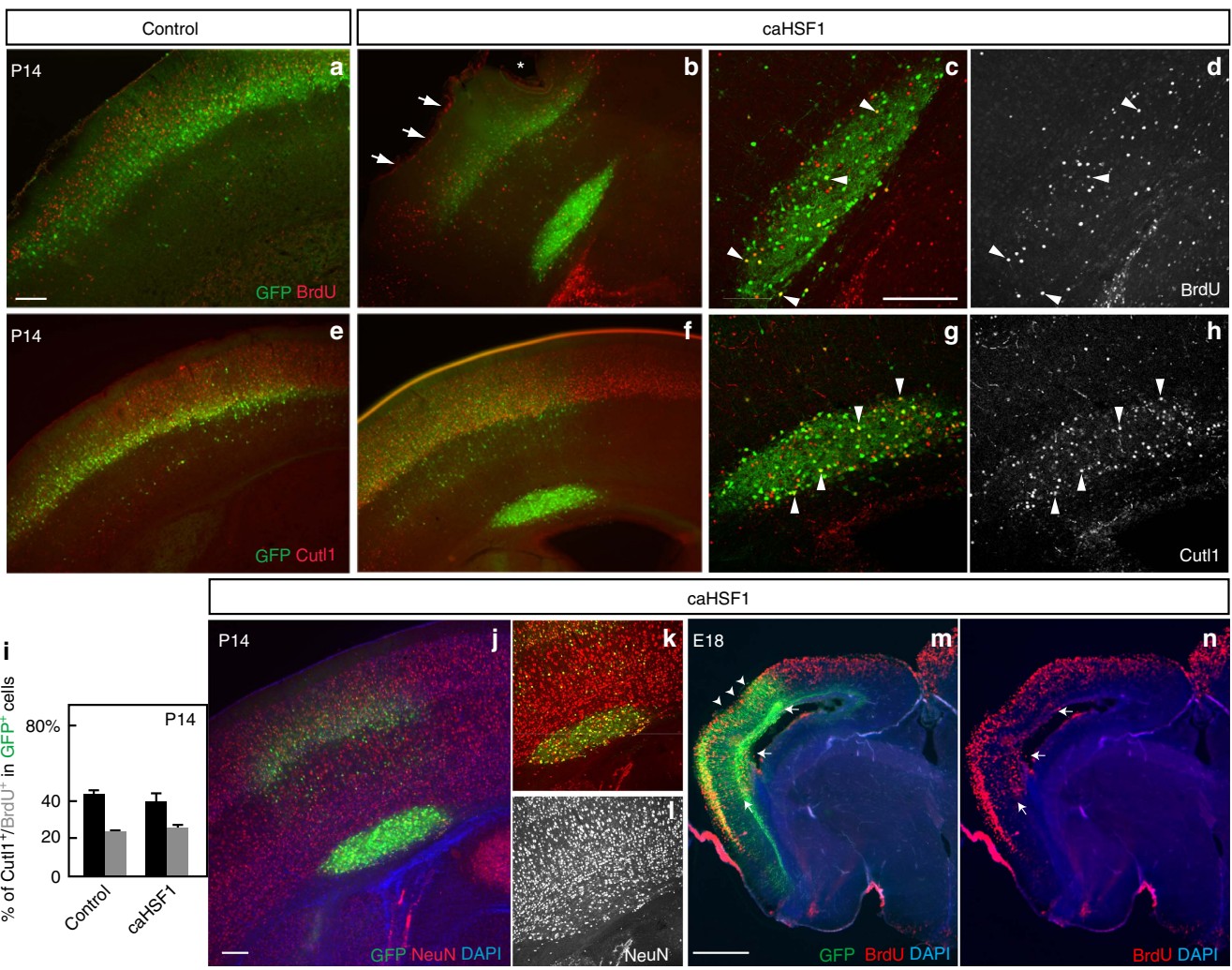

**Figure 6 | caHSF1 disturbs neuronal positioning without affecting differentiation.** (**a**–**h**) Mice were electroporated with pCAGIG (control) or pCAGIG-caHSF1 at E14 and stained for BrdU (injected at E15) or Cutl1 (a marker for upper layer neurons) (red or white) at P14. caHSF1-electroporated (GFP$^+$) cells in the heterotopia include Cutl1$^+$ and BrdU$^+$ cells. Most non-electroporated BrdU$^+$ neurons were localized in upper layers (layers II-IV). Arrows and an asterisk indicate disruptions in the marginal zone and pial surface above the heterotopia (**b**). Arrowheads indicate representative double-labelled cells (**c,d,g,h**). (**i**) The percentages of Cutl1$^+$ (black) and BrdU$^+$ (gray) cells within caHSF1-electroporated cells are comparable to those within control-electroporated cells, indicating that caHSF1 does not affect differentiation. Data are represented as mean ± s.e.m. $P$ = n.s. by t-test ($n$ = 5 each). (**j**–**l**) Immunostaining for NeuN (a marker for mature neurons, red or white) on the P14 cortex shows normal neuronal differentiation of caHSF1-electroporated (at E14) cells (labelled with GFP) even within the heterotopia. (**m,n**) BrdU$^+$ caHSF1-electroporated (at E14) cells (GFP$^+$) were found in deep layers at E18 (arrows). BrdU was incorporated at E15. Disruption in the cortical surface (arrowheads) was already found at E18 in some caHSF1-electroporated samples. Scale bars = 0.2 (**a**-**l**) and 0.5 (**m,n**) mm.

experiments demonstrate the stochastic response of Hsf1–Hsp signalling to EtOH or other exposure among homogeneous cell populations (Fig. 1, Supplementary Figs 1-2). Therefore, we believe that these various patterns of Hsf1–Hsp activation in tissue/organs strongly depend on, though may not be entirely due to, the stochastic response within the same cell populations. Reporter expression turnover, for example, may be an additional potential factor for the cell-to-cell difference. Given the unmet need for proper targeted therapies following acute exposure to environmental stressors during pregnancy, understanding the mechanisms of the stochastic activation and potential buffering mechanisms of such stochasticity is a critical next step. Mechanisms similar to TATA-box-involved transcriptional controls for evolutionary adaptation in microbes[17,37] may be involved in the generation of stochasticity among cells. Differential cell fates of NPCs (neuronal versus glial, inhibitory versus

excitatory neurons and so on) are additional factors that may affect cell-to-cell variability. A buffering mechanism of unequal transcriptions at the post-transcription level has recently been reported[38].

We have shown that stochastic activation of Hsf1–Hsp signalling is induced *in vitro* by EtOH, $H_2O_2$ and MeHg (Fig. 1), as well as *in vivo* by prenatal exposure to EtOH using our reporter mice (Fig. 3). Hsf1–Hsp activation also occurs in the brain in a stochastic manner after prenatal exposure to several other stressors including MeHg, suramin, maternal seizures and sodium arsenite[22]. In addition, prenatal exposure to suramin, similar to EtOH (Fig. 3), is known to cause neuronal migration defects[22]. Therefore, these stressors are likely to share specific mechanisms presented in this study to ultimately affect cortical development. One potential mediator may be oxidative stress; many environmental factors including EtOH causes

oxidative stress[39,40], and oxidation of HSF1 has been suggested to enhance HSF1 DNA binding activity to induce Hsf1–Hsp signalling[41].

Several questions remain to be answered, including is whether Hsf1–Hsp signalling and other stress responsive signalling pathways are functionally integrated; whether their activations

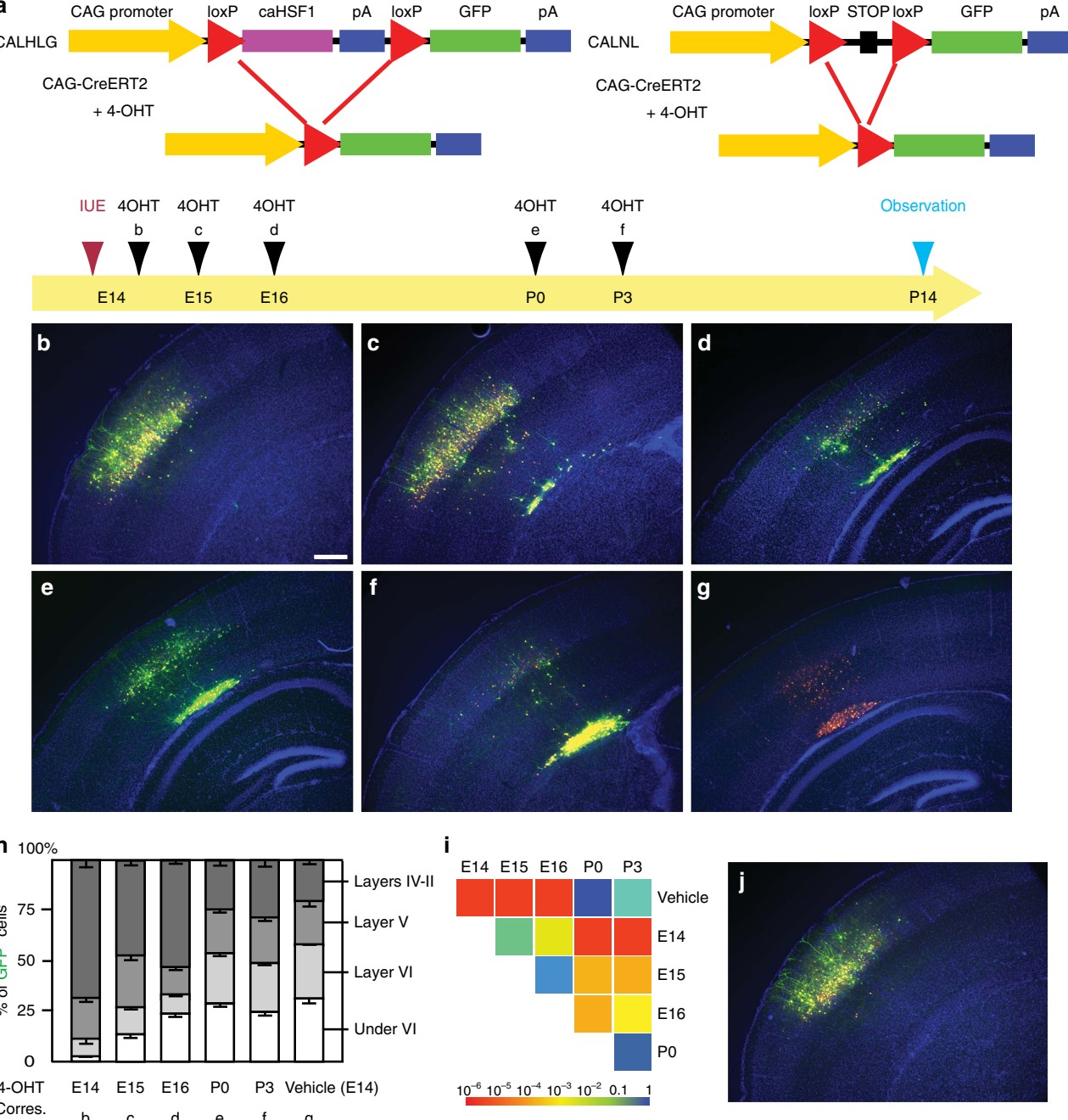

**Figure 7 | Heterotopias formed by temporal prenatal expression of caHSF1 persist after birth. (a)** Scheme describing the experiments. CAG-promoter-loxP-FLAGcaHSF1-loxp-GFP (CALHLG) with CAG-RFP and CAG-CreERT2 was electroporated at E14. 4-OHT was administered at the indicated time points, and all samples were fixed at P14. 4-OHT administration at any time point consistently resulted in excision of the FLAGcaHSF1 unit (indicated by GFP expression) in over 95% of RFP$^+$ electroporated cells. FLAG immunohistochemistry confirmed complete sequestration of FLAG-tagged caHSF1 by 2 days after 4-OHT injection. **(b–i)** Distribution of the electroporated cells was observed at P14 after administration of 4-OHT or vehicle only (control) at different time points **(b–g)**, and quantitatively analysed **(h,i)**. Scale bar, 0.5 mm. Panel **i** shows pairwise comparisons of cumulative radial distributions using Kolmogorov–Smirnov (K–S) test. Each P value is shown in a color according to the scale at the bottom. Without 4-OHT (vehicle only), CALHLG expression causes heterotopia formation **(g,h)**, similar to pCAG-caHSF1 introduction. Delay of caHSF1 until after birth (4-OHT injection at P0 or P3) causes the formation of large heterotopias **(e,f,h)** (**i**: P = NS, K–S test). Expression of caHSF1 at low levels for a very short period (4-OHT injection 6 h after IUE at E14), did not form heterotopias **(b,h)** (P = NS compared with pCALNL-GFP, n = 6). Expression of caHSF1 until E15 or E16 was sufficient for heterotopia formation **(c,d,h)** (P < 0.05 compared with the vehicle control or pCALNL-GFP (see **j**), n = 13, 5, respectively). Data are represented as mean ± s.e.m. **(j)** No migration defects by electroporation of pCALNL-GFP with CAG-CreERT2 and pCAG-RFP followed by 4-OHT injection at E16 (see **a**).

are concomitantly regulated, or if they take part in overlapping or compensatory roles; and how excessive Hsf1–Hsp activation delays the migration of cortical cells. We did not observe the delay of neuronal migration in either the *Hsf1* KO cortices even when under EtOH exposure[15]. The overexpression of *Hsp70* did not affect the migration of cortical cells (Supplementary Fig. 7b), suggesting that the impaired neuronal migration may be mediated by non-canonical downstream targets of Hsf1. One possibility mechanism may be through controlling the transcription of microtubule-associated molecules such as *Dcx*, *Dclk*, *p35* and *Ndel1* via interaction with Hsf2 (ref. 42). Of note, the cortical neurons that have stalled due to the lack of *Dcx* re-migrate by replenishment of the gene after birth[43] while the reduction of Hsf1 activity after birth is not sufficient to reboot this migration (Fig. 7). This suggests that genes other than microtubule-associated genes may critically mediate the migrational disruptions resulting from high-level Hsf1 activation. Other potential downstream target genes may be the genes

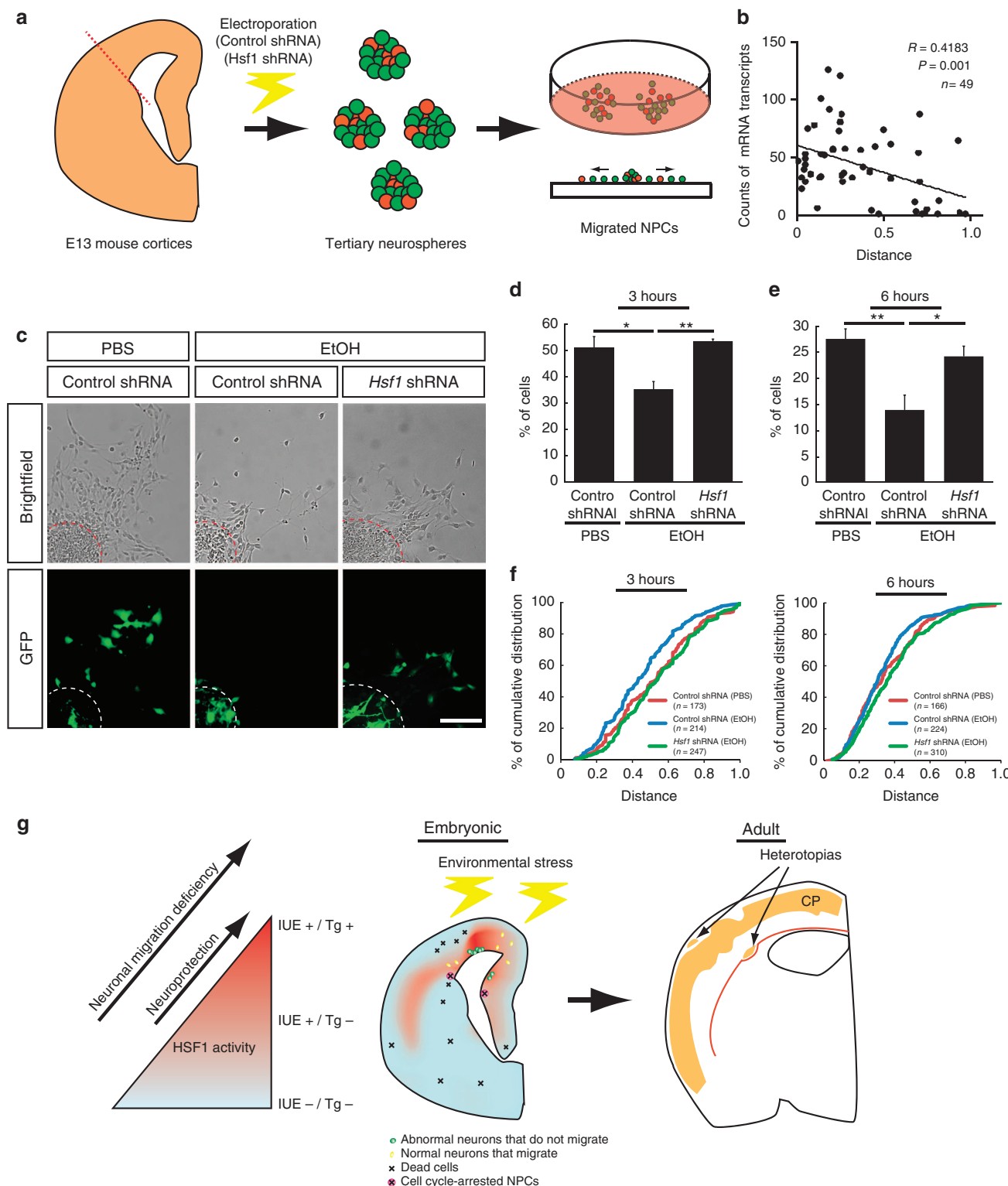

controlling adherens junction formation which are disturbed by excess activation of Hsf1 (Fig. 5).

## Methods

**hNPC culture.** Induction of the hNPCs from iPSCs (AG09429 [female, 25 years of age] and GM03440 [male, 20 years of age]) has been described previously[44]. hNPCs were maintained on poly-L-ornithine/laminin-coated plates in DMEM/F12 medium (Life Technologies) containing N2 and B27 supplements (Life Technologies), laminin (Life Technologies) and FGF2 (Peprotech). hNPCs were passaged once a week after dissociation with TrypLE Select (Life Technologies). For analysis, hNPCs were plated on poly-L-ornithine/laminin-coated glass-bottom dishes (14 mm diameter No. 0 coverglass in 35 mm petri dish) (MatTek Corporation).

**smFISH.** smFISH was performed as previously described[45,46] with minor modifications. Briefly, hNPCs were fixed with 100% EtOH (Sigma) for 10 min in 1× PBS solution at room temperature. Following fixation, the cells were washed twice with 1× phosphate buffer and then permeabilized with 70% EtOH. Cells were then washed with wash buffer containing 10% formamide (Ambion) and 2× saline sodium citrate (SSC) (Sigma). The cells were incubated with smFISH probes in hybridization buffer consisting of 10% formamide, 2× SSC, and 10% dextran sulfate (Sigma) at 37 °C overnight in a dark and humidified chamber. The probes for both human and mouse *Hsp70* were custom designed (Supplementary Table 2). Predesigned probes for *Gapdh* were purchased from Biosearch Technologies, Inc. Following hybridization, the samples were washed with wash buffer for 30 min at 37 °C, counterstained with DAPI (1:2,000), mounted with VECTASHIELD (VECTOR laboratories), and imaged using a Zeiss LSM510 confocal microscope. For the combination of smFISH analysis and Ki67 staining, which was used to distinguish between G1-S and G2 phases of the cell cycle[47,48] the cells were incubated with anti-Ki67 (1:10, Clone B56, BD Biosciences, 550609) in addition to smFISH probes (human *HSP70*, 1:100). The data were analysed using semi-automated software for counting mRNA particles (StarSearch; http://rajlab.seas.upenn.edu/StarSearch/launch.html).

**Slice culture of human fetal brains.** GW15-18 human fetal brains were collected according to guidelines on the research use of human brain tissue from the New York State-licensed Human Fetal Tissue Repository at the Albert Einstein College of Medicine. This study was approved by the Human Investigation Committees at Children's National Medical Center. The cerebral cortex was collected in cold Hibernate media + B27 supplement (Gibco), separated from surrounding tissue, and immediately dissected into 1 cm-thick blocks. Subsequently, the tissue blocks were cut with a tissue chopper at 0.3 mm thickness. The slices were cultured with or without 50 mM EtOH for 24 to 72 h on a Millipore membrane. After the fixation with 4% paraformaldehyde, the slices were cut with a cryostat. A part of the tested tissues was stored before the culture. The intactness of the tissue and the amount of prenatal environmental stress were assessed by histological staining (hematoxylin and eosin and TUNEL) and qRT–PCR (for stress, inflammation and cell death markers, as well as growth factors)[49]. Samples that were judged as 'not fresh' or 'damaged' were excluded from the study.

**Animals.** Animals were handled according to protocols approved by the Institutional Animal Care and Use Committee of Children's National Medical Center on 05/04/2015 (The protocol number is 00030323). Generation and genotyping of *Hsf1* KO mice are described[50]. The HSE–RFP fragment was purified according to the standard protocol for microinjection. Microinjection into C57BL/6J X SJL/J was performed by Animal Genomics Services at Yale University. The founder lines were screened by PCR genotyping, and RFP expression was confirmed in the neonates from EtOH-treated dams under a dissecting microscope equipped with epifluorescence. Of the 3 founder lines obtained, we selected the most sensitive reporter line, which was then backcrossed 7 times with C57BL/6J mice. The embryos or pups from C57BL/6 wild-type females crossed with HSE–RFP homozygous transgenic males were used for the study. For routine genotyping, the following oligonucleotide PCR primers were used: Forward 5′-AAGGTGTACGTGAAGCACCC-3′, Reverse 5′-CCCATGGTCTTCTTCTG CAT-3′, to amplify a 250 bp band of a partial *DsRed2* gene. Hotstar Taq DNA polymerase kit (Qiagen) was used.

**Whole-genome sequencing of HSE–RFP transgenic mouse.** With the genomic DNA extracted from transgenic mice, we performed shallow whole genome sequencing (WGS) using Illumina HiSeq 2500 at 101 bp paired-end at ~3× whole-genome coverage[51]. To identify candidate integration sites, we aligned paired-end reads to the reference mouse genome using Bowtie2 (ref. 52) in global (reads align end to end) and local (reads are soft-clipped) modes allowing 0 or 1 mismatches for potential SNPs or PCR errors[53,54], and reported discordant read pairs. As an alternative approach, we also ran BWA using the BWA-MEM algorithm, which performs local alignment and supports split (chimeric) read alignments, in which the aligner assumes a read and is mapped to two separate locations because of possible structural variation[54,55].

**Drug administration.** Pregnant mice received intraperitoneal injections of 25% EtOH at 2.0 g kg⁻¹ daily at E14-16. This regimen does not induce the migration deficits in most of the regions in the cerebral cortex[15]. Cortical slices obtained from GW 15–18 human fetuses were cultured for 24 h at 50 mM EtOH in Neurobasal medium with B27 supplement and 5% FBS. The mouse had a peak Blood Alcohol Concentration (BAC) around 200–250 mg dl⁻¹ within a half hour after an administration. This falls in a range between the consumption levels by social drinkers (60 mg dl⁻¹) and those by chronic alcoholics (320–620 mg dl⁻¹)[56,57], and is also similar to that used in previous studies reporting pathological and molecular effects of fetal alcohol exposure[58,59]. The human *in vitro* model used the same range of EtOH concentration (50 mM (230 mg dl⁻¹)), by which several aspects of FASD phenotypes have been shown to be induced in the mouse cerebral cortex *in vitro*[60]. Alcohol concentration was stable for 24 h in culture[61]. Consecutive brain slices were used for control and alcohol conditions to minimize regional differences between samples. At least three pairs of slices were used for evaluation of the results in each condition.

**Immunohistochemistry.** Immunohistochemistry was performed principally by the following methods previously described[27]. Sections were pre-treated with Proteinase K before TUNEL staining using ApopTag peroxidase in situ apoptosis detection kit (Millipore). The staining was amplified using biotin-conjugated secondary antibodies, VECTASTAIN ABC system (Vector) and TSA Plus system (Perkin Elmer). Monoclonal anti-BrdU (1:100, Clone 3D4, BD Biosciences, 555627), anti-Calbindin (1:1000, Abcam, ab82812), anti-Tuj1 (1:1,000, Abcam, ab78078), anti-Nestin (1:50, rat-401, DSHB), anti-FLAG M2 (1:50, Sigma, F1804) and anti-NeuN (1:500, Clone A60, Millipore, MAB377) antibodies were used. Polyclonal anti-HSF1X[50] (1:500), anti-HSF1 antibody (1:200, C-19, SCBT, sc-8061) and anti-Cutl1 (1:100, M-222, SCBT, sc-13024) antibodies were also used. The HSF1 expression pattern in Supplementary Fig. 5 was consistently obtained by using both HSF1X and HSF1 C-19 antibodies.

**qRT–PCR.** The cells were first dissociated with TripLE Select (Invitrogen) from the dissected cortex, and were sorted using FACS Vantage SE. Total RNA was isolated using the RNeasy Plus kit (QIAGEN), and the quality of the total RNA was evaluated by Bioanalyzer RNA 6000 kit (Agilent). Samples showing RNA integrity number >9 were used. cDNA was synthesized by using SuperScript First-strand synthesis system for RT–PCR with random hexamer primers (Invitrogen).

**Figure 8 | Reducing excessive Hsf1 activation mitigates EtOH-induced migration deficits.** (**a**) Schematic drawing of the migration assay. (**b**) The levels of *Hsp70* expression (smFISH) are negatively correlated with the migration distance of cells from the center of the neurosphere after 3-hour culture. The position of the cell that had migrated the longest distance from the center (0.0) was set as 1.0, and was used for determining the relative positions of the other cells. (**c**) Representative images of the migration assays after a 6-h culture. Scale bar, 0.1 mm. (**d,e**) The percentages of GFP⁺ cells within the outer half of the maximum migration distance from the center of the neurosphere in total GFP⁺ cells. Four independent experiments were performed per group. Data are represented as mean ± s.e.m. One-way ANOVA, F(2,9) = 9.15, P = 0.007 (d), F(2,9) = 9.59 P = 0.006 (e). *P < 0.05, **P < 0.01 by post hoc Tukey test. (**f**) Cumulative distribution of the cells shows the alleviation of EtOH-induced migration deficits by *Hsf1* shRNA (n > 150 cells from 4 biological replicates per group, P < 0.05 by K-S test for all pairs except the pair of Control shRNA (PBS) (red) versus *Hsf1* shRNA (EtOH)(green). (**g**) A model for the production of various cortical dysplasia due to probabilistic Hsf1–Hsp signalling activation elicited by prenatal environmental stressors. Excessive activation of HSF1 in a subpopulation of cortical cells, as detected in the reporter transgenic mouse (left), disrupts their normal developmental processes such as migration (green cells in middle panel). Moderate activation of the signalling that is detectable by *IUE*-based method, but not in the transgenic mouse (Fig. 3), serves to protect the cells from the environmental stressors[15] (Fig. 2, left), and allows the cells develop normally (yellow cells in middle). Meanwhile, another subpopulation of cells may not have enough HSF1 activation to protect them from damage elicited by environmental challenges, thereby resulting in focal accumulation of dead cells or NPCs with impaired proliferation[15] (Fig. 2, the cells indicated by X in middle panel). These heterogeneous events of abnormal development occur probabilistically (Fig. 1), accounting for individually distinct pattern of focal cortical malformations in the cortex exposed to similar levels of environmental challenges (right panel).

GAPDH levels were detected by Taqman rodent GAPDH control reagents and used for normalization. Thermocycling was carried out by using the Applied Biosystems 7900 system and monitored by SYBR Green I dye detection. For qRT–PCR of mHsp70; Forward 5′-ggccagggctggattact-3′ and Reverse 5′-gcaacca ccatgcaagatta-3′ primers were used.

**Flow cytometric analysis.** The cortex was dissected into small pieces in cold PBS; these pieces were then incubated with TripLE Select (Invitrogen) at 37 °C for 10 min. After dissociation, cells were immunostained with PBS containing 0.5% BSA and 0.1% NaN3. LIVE-DEAD Fixable Far Red Dead Cell Stain Kit for 633 or 635 nm (Thermo Fisher Scientific) was used for the staining of dead cells. Using Perm & Fix (Invitrogen), the cells were stained using RFP-biotin (Abcam), NeuN-FITC (Millipore) or Tuj1-FITC (eBioscience) (all used at 1:500). Streptavidin-PE (eBioscience) was used at 1:2000. Antibody reaction was done for 30 min on ice. The recording and analysis were performed using BD LSRII with Green laser and DIVA software, and FlowJo (TreeStar), respectively. A total of approximately 100,000 cells were recorded per sample.

***In utero*** **electroporation and 4-OHT injection.** *In utero* electroporation was performed at E14 as previously described[62–64]. The plasmids used were: pCAG–GFP (0.5 mg ml$^{-1}$), pCAG-RFP (0.5 mg ml$^{-1}$), pCALSL-RFP (2 mg ml$^{-1}$), pCALSL-caHSF1 (2 mg ml$^{-1}$), pTα1-Cre (1 mg ml$^{-1}$), pCALHLG (2 mg ml$^{-1}$), pCALNL-GFP (2 mg ml$^{-1}$), shRNA clones (2 mg ml$^{-1}$), pCAGIG-caHSF1, Hsf1WT, Hsf1R71A (2 mg ml$^{-1}$), pCAGIG-HSP70 (2-5 mg ml$^{-1}$), pCAG-CreERT2 (1 mg ml$^{-1}$). Control experiments were performed using empty vectors or control shRNA constructs at corresponding concentrations. For some constructs, gene expression was temporally controlled by 4-OHT (Sigma). 4-OHT was diluted in corn oil at a concentration of 20 mg ml$^{-1}$, and administered through intraperitoneal injections (2 mg per 100 g body weight). The same solution without 4-OHT was used as a vehicle control.

**Plasmid construction.** Using the Vista Genome Browser (http://pipeline.lbl.gov/cgi-bin/gateway2), we predicted that the Hsp70 promoter region is conserved across mammalian species. The sequence of this region was amplified by PCR from a mouse BAC clone that includes Hsp70 gene and the surrounding genomic region (BACPAC). This was confirmed through sequencing (Keck laboratory at Yale University) that the obtained 649 bp fragment contains two HSF1 binding sites (HSE (heat shock-responsive element), Supplementary Fig.3). The PCR product was inserted into pDsRed2-1 (Clontech) to generate the pHSE–RFP reporter plasmid. The mutant HSE (mutHSE), which contains 14 point mutations that prevent Hsf1 binding to HSE, was generated using the Quickchange site-directed mutagenesis kit (Stratagene). pCAGIG-caHSF1 was constructed by inserting caHSF1 which is the ΔRDT mutant[65] into the pCAGIG plasmid which contains IRES-GFP. HSP70 and a series of HSF1 mutants were inserted into pCAGIG (Addgene). shRNA of Hsf1; C2 (ref. 66) (Sigma), SA4 (ref. 15) (Superarray), and their control shRNA were purchased. Construction of pTα1-Cre and pCALSL-RFP were reported previously[27]. pCALSL-caHSF1 was constructed by inserting FLAG-caHSF1 and polyA from the pCDNA3.1 construct[67]. In the samples co-electroporated with pCALSL-RFP, pCALSL-caHSF1 and pTα1-Cre, FLAG immunostaining co-localized with RFP, confirming RFP as a marker for caHSF1-expressing cells. pCALHLG was generated from pCALNL-GFP (Addgene) by replacing Neo/Kan to caHSF1. pCAG-CreERT2 was purchased (Addgene).

**Immuno-electron microscopic analysis.** Serial sections of tissues were made using a standard procedure in our laboratory[68], and processed for the digital camera system[25,69]. Briefly, post-fixed tissue after immunohistochemistry with GFP antibody was embedded in Durcupan on microscope slides and coverslipped. Selected immunoreactive cells were traced with Neurolucida and photographed with an Axioplan 2 microscope (Zeiss). Selected areas were re-embedded into Durcupan blocks and cut by a Reichert ultramicrotome into 70 nm thick sections. These sections were then stained and imaged in a JEOL 1010 electron microscope equipped with Multiscan 792 digital camera (Gatan, Pleasanton, CA).

***In vitro*** **migration assay.** E13 mouse cortices were dissected, and the NPCs were dissociated for neurosphere suspension culture. The NPCs were cultured at 1.0 × 10$^5$ cells ml$^{-1}$ in NPC medium[44] in the presence of 20 ng ml$^{-1}$ EGF (Peprotech) and 20 ng ml$^{-1}$ FGF-2 (Peprotech) for 7 days. The culture was expanded by passaging three times. For electroporation, 8 μg of control shRNA or Hsf1 shRNA[15], 4 μg of Hsf1 shRNA and full-length human HSF1 and 2 μg of pMAX-GFP (Amaxa) were used for 2–4 × 10$^6$ NPCs dissociated from the neurospheres. After electroporation, the NPCs were cultured with non-electroporated NPCs for 3 days to form neurospheres. The formed neurospheres with a diameter of 50-100 μm were then manually placed onto coated glass-bottom dishes (MatTek Corporation), and cultured either with 400 mM EtOH or PBS for 3 or 6 h. After fixation, migrated NPCs were counted. The cells were also immunolabelled with anti-Nestin (1:50, DSHB), anti-Ki67 (1:10, Clone

B56, BD Biosciences, 550609), and anti-cleaved Caspase-3 (1:100, Cell Signaling Technology, #9654) antibodies.

**Statistics.** Data were subjected to statistical analyses with SPSS (IBM), Prism 6 (GraphPad), and SigmaPlot 12 (Systat Software). Results were expressed as mean ± s.e.m. unless otherwise indicated. All statistical analyses and calculation of P values were performed using either two-tailed Student's *t*-test for pairwise comparisons, 1-way ANOVA, followed by Tukey's multiple comparison tests for multiple group comparisons, or Kolmogorov-Smirnov (K-S) test for the equality of probability distributions between two samples. Grouped analyses were completed without repeated measures. Non-normally distributed data were analysed by the corresponding nonparametric tests. Sample size was described in the legend of each figure. The variance was similar between groups being statistically compared. The sample size was at least three in most cases, and it allows us to achieve at least 80% standard power to detect the difference with 95% confidence. The sample size was not predetermined by statistical method. No data points were excluded from the analysis. We used blinded outcome assessment for our animal studies. All data collection and analyses were performed by experimenters blind to the experimental conditions (genotypes, treatments etc.). A P value of less than 0.05 was considered significant. For statistical testing of probabilisticity, an experimental data set of predetermined 18 cell positions from each dish of hNPCs exposed to three types of environmental stress (12 dishes per type) was shuffled to generate a permutated set, and the average of Pearson Correlation Coefficients (PCC) within the set was calculated using MATLAB. This process was repeated one thousand times to generate a distribution of the average PCC for the permutated sets, which was then compared with the average PCC within the experimental data set by employing one-sample Z-test[70].

**Data availability.** The data that support the findings of this study are available from the corresponding author on request.

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

## Acknowledgements

We thank Sho Yamada, Miki Masuda, and Hiroki Naito for their technical assistance, and Dr. Aminah Sheikh for critical reading of the manuscript. VM laboratory has been supported by the Agence Nationale pour la Recherche (Program SAMENTA

ANR-13-SAMA-0008-01), Fondation Jérôme Lejeune, and IREB/FRA (2014/18). This work was supported by a NIH/NIAAA (R00AA01838705, R01AA025215), CTSI-CN Pilot Research Award and Avery Translational Research Career Development Program Award from Children's National Medical Center, Brain and Behavior Research Foundation (Scott-Gentle Foundation and Essel Foundation), ABMRF, and the Kavli Institute for Neuroscience at Yale.

## Author contributions

K.H.-T. and P.R. conceived the project. Y.M., A.S., M.R., M.F., M.T., S.I., K.B. and K.H.-T. performed experiments. Analyses were performed by S.I. M.T. and K.H-T. S.I., A.S., M.T., F.G., K.B., P.R. and K.H.-T. wrote the manuscript with inputs from A.N. and V.M.

## Additional information

**Competing interests:** The authors declare no competing financial interests.

