## [Peer Review File · Nature Communications]

Reviewers' comments:

Reviewer #1 (Remarks to the Author):

In their manuscript Ishii et al., address important aspects relating to the effects of prenatal stress on brain development. In their study, Ishii et al., found that the primary stress response transcription factor Hsf1 is upregulated to highly variable degrees and in a stochastic manner in mouse and human embryonic tissues and hNPC cell lines, upon environmental stressors. The authors then evaluated the consequences upon high expression levels of Hsf1 and discovered that excessive upregulation of Hsf1 affects neuronal migration cell-autonomously during cortical development. Furthermore, non-cell-autonomous and/or 'community' effects also contribute to disturbances of embryonic brain development and heterotopia formation upon excess Hsf1 activation. The authors find that even a short period of Hsf1 overactivation during prenatal development results in migration deficits, and that the duration of Hsf1 overactivation influences the severity of the defects. Altogether the authors, based on their findings, suggest that stochastic activation of Hsf1 at high levels may reflect the underlying cause of the heterogeneous nature of focal cortical malformation in response to prenatal environmental stressor(s).

In general, Ishii et al., address an important and very timely question. Their work is likely to be of great interest to a larger readership and certainly also beyond the neurodevelopmental field. The authors use highly innovative methodology to address the functional consequences and phenotypic manifestations upon Hsf1 overactivation in the developing brain, which occurs upon exposure to environmental stressors. Most of the data is of high quality and all results reported appear highly significant as illustrated in the figures. The authors' conclusions and interpretation of their results seem reasonable, establish an interesting hypothesis and open a variety of interesting new directions and questions for future research.

Altogether, this study is well executed, the topic is important and timely. The manuscript is clearly written, the experiments well performed and the findings illustrated nicely. Some of the claimed findings could however be further strengthened by addressing the following points:

1. It would be helpful to present real high resolution (single cell level) smFISH images in Figure 1 to better illustrate the stochasticity/heterogeneous expression of HSP70 at the qualitative level. Instead, the 'secondary data' panels from the different hNPC line could be moved to supplemental data.
2. The results regarding dying cells in Figure 2j-k are significant with a p value of 0,0262. It would be valuable however to show apoptotic cells using TUNEL and/or caspase-3 stainings.
3. While the IUE-mediated transfer of reporter constructs (pHSE-RFP) and obtained data is very convincing, the HSE-RFP transgenic reporter lines are somewhat suboptimal. It is assumed that these transgenic lines include founders with random insertion of the reporter cassette? If yes, can the authors exclude that heterogeneous expression of the reporter (upon induction by EtOH) is also occurring in combination with local regulatory elements near the insertion site? The authors would like to provide information regarding these transgenes in a separate manuscript, which is fine, but it would be useful to get at least some information regarding the transgene insertion site. Can the authors be sure that reporter gene expression could in principle occur ubiquitously, but that the stochastic expression observed is entirely due to the stochastic response of neural progenitors to EtOH exposure? Reporter expression seems to also occur in the eye (Figure 3b) and other parts of the embryo. What is the author's interpretation of these results regarding cell-type specificity of the stochasticity of cellular response to EtOH exposure? Clarifying the above issues will significantly strengthen the interpretation of the results obtained from these transgenic mice.

4. The authors give appropriate credit to previous work. However, the introduction and discussion could be extended to provide a bit more context, background and interpretation of the wealth of data presented.

Reviewer #2 (Remarks to the Author):

The manuscript by Ishii et al reported that overactivation of Hsf1-Hsp signaling disturbed neuronal migration in the cerebral cortex which is in part a pathological mechanism underlying abnormalities in brain formation caused by developmental alcohol exposure. As the author stated, they previously reported reduction of Hsf1-Hsp signaling activity also induce developmental abnormalities. Over all, the authors successfully demonstrated discordant Hsf1-Hsp signaling play critical for the effect of environmental stress on brain development. The manuscript is very well written, but there are several issues need to be addressed.

The title of the current manuscript is too vague. Since this is research article, not review article, need modify it more specific (e.g., what signaling pathway, what prenatal stress) based on their findings in this study.

Abstract should be re-written with more specific and conservative tone. Although the author tested several environmental stresses in Figure 1, most of the data were obtained in EtOH model. It is still unclear that specific mechanisms the authors presented here is also involved in developmental phenotypes caused by other stressors. The authors should also specify EtOH as a stressor (i.e., EtOH-induced Hsf1 overactivation) in result sections.

Figure 3b, please show the comparative images of cellular distribution of Tuj1 neurons in EtOH-treated and PBS-treated mice. Only image of EtOH-treated mice is available in the current manuscript.

In Figure 3g''', the authors successfully demonstrated delayed migration associated with HSP activity. Please present representative images of each condition with bin information. The distribution of RFP-positive cells looks very heterogeneous. How did the authors determine region of interest for quantitative analysis? There is no information of bin analysis for neuronal migration with ROI information in the method section which should be corrected.

Figure 5c

Distribution of GFP-positive cells in control treated with EtOH does not seem to be different compared with those in control without EtOH. Does EtOH have no effect on neuronal migration?

Figure 6 b', b'', d', d''

In the graph presented in Fig 6e, approximately 40% and 20% of GFP-positive cells are Cutl1-positive and BrdU-positive respectively. However, a very few subpopulation of GFP-positive cells looks Cutl1- and Brdu-positive, which is not consistent with the quantitative results. Please replace them to more representative images.

In Figure 8, the author elegantly demonstrated importance of Hsf1 signaling pathway in neuronal migration defects caused by EtOH treatment. This is the key data supporting the authors' mechanistic hypothesis. One major concern is that the author did not demonstrate if overexpression of Hsf1 normalizes the effect of Hsf1 shRNA. Given that off-target effects are always concerns in shRNA experiments, the authors should test it to confirm that activation of Hsf1 is a mechanism of EtOH-induced migration defect.

Minor points;

Page 8 line 169: Did the author use DsRed2 protein? If not, please correct it.

To the Reviewers:

We greatly appreciate the input by the reviewers and their enthusiasm for our work. The criticisms presented have been valid, constructive, and have produced a far stronger manuscript. We have addressed the concerns of the reviewers as follows:

Reviewer #1:

In their manuscript Ishii et al., address important aspects relating to the effects of prenatal stress on brain development. In their study, Ishii et al., found that the primary stress response transcription factor Hsf1 is upregulated to highly variable degrees and in a stochastic manner in mouse and human embryonic tissues and hNPC cell lines, upon environmental stressors. The authors then evaluated the consequences upon high expression levels of Hsf1 and discovered that excessive upregulation of Hsf1 affects neuronal migration cell-autonomously during cortical development. Furthermore, non-cell-autonomous and/or 'community' effects also contribute to disturbances of embryonic brain development and heterotopia formation upon excess Hsf1 activation. The authors find that even a short period of Hsf1 overactivation during prenatal development results in migration deficits, and that the duration of Hsf1 overactivation influences the severity of the defects. Altogether the authors, based on their findings, suggest that stochastic activation of Hsf1 at high levels may reflect the underlying cause of the heterogeneous nature of focal cortical malformation in response to prenatal environmental stressor(s).

In general, Ishii et al., address an important and very timely question. Their work is likely to be of great interest to a larger readership and certainly also beyond the neurodevelopmental field. The authors use highly innovative methodology to address the functional consequences and phenotypic manifestations upon Hsf1 overactivation in the developing brain, which occurs upon exposure to environmental stressors. Most of the data is of high quality and all results reported appear highly significant as illustrated in the figures. The authors' conclusions and interpretation of their results seem reasonable, establish an interesting hypothesis and open a variety of interesting new directions and questions for future research.

Altogether, this study is well executed, the topic is important and timely. The manuscript is clearly written, the experiments well performed and the findings illustrated nicely. Some of the claimed findings could however be further strengthened by addressing the following points:

We appreciate the reviewer's important suggestions. All comments are valid, and have greatly improved our manuscript.

1. It would be helpful to present real high resolution (single cell level) smFISH images in Figure 1 to better illustrate the stochasticity/heterogenous expression of HSP70 at the qualitative level. Instead, the 'secondary data' panels from the different hNPC line could be moved to supplemental data.

We appreciate this reviewer's comment. We have changed Figure 1 and created a new Supplementary Figure 1 accordingly.

2. The results regarding dying cells in Figure 2j-k are significant with a p value of 0,0262. It would be valuable however to show apoptotic cells using TUNEL and/or caspase-3 stainings.

We agree and now have added images of caspase-3 staining (Fig.2p,p').

3. While the IUE-mediated transfer of reporter constructs (pHSE-RFP) and obtained data is very convincing, the HSE-RFP transgenic reporter lines are somewhat suboptimal. It is assumed that these transgenic lines include founders with random insertion of the reporter cassette? If yes, can the authors exclude that heterogenous expression of the reporter (upon induction by EtOH) is also occurring in combination with local regulatory elements near the insertion site? The authors would like to provide information regarding these transgenes in a separate manuscript, which is fine, but it would be useful to get at least some information regarding the transgene insertion site. Can the authors be sure that reporter gene expression could in principle occur ubiquitously, but that the stochastic expression observed is entirely due to the stochastic response of neural progenitors to EtOH exposure? Reporter expression seems to also occur in the eye (Figure 3b) and other parts of the embryo. What is the author's interpretation of these results regarding cell-type specificity of the stochasticity of cellular response to EtOH exposure? Clarifying the above issues will significantly strengthen the interpretation of the results obtained from these transgenic mice.

We appreciate the reviewer's suggestion and have now included data of transgene insertion sites obtained by NGS whole genome sequencing of reporter transgenic mice (Supplementary Figure 4). These data indicate that the local cis genomic regions are unlikely to affect the expressions of the transgenes.

The interpretation of the stochasticity and potential confounders is now discussed extensively in the Discussion (page 15-16). Regarding the ubiquity of reporter gene availability, our newly added whole genome sequencing of the reporter transgenic mice indicates the reporter expression to occur ubiquitously throughout the body. We did observe broad expression in many tissues/organs including eye, heart, liver, and skin¹ supporting this. Expression patterns across various tissues and organs vary between animals. Potential factors for these differences include differential circulation/accumulation of EtOH between tissues and organs, differential activation of HSF1-HSP signaling between tissue/organ-specific cell types, and stochastic response of cells across tissue/organs.

Regarding the stochasticity, the reporter expression in each tissue/organ always shows stochastic patterns in response to the stressors as far as we observed. The results of our *in vitro* experiments demonstrate the stochastic response of the HSF1-HSP signaling to EtOH or other exposure among homogeneous cell populations (Fig. 1, Supplementary Fig.1-2). Therefore, we believe that these patterns of HSF1-HSP activation in tissue/organs also strongly depend on (though may not entirely due to) the stochastic response within the same cellular populations. For example, reporter expression turnover may be an additional potential factor that induces the

cell-to-cell difference. Our interpretation described here is now included in the discussion (page 15-16, line 312-323).

4. The authors give appropriate credit to previous work. However, the introduction and discussion could be extended to provide a bit more context, background and interpretation of the wealth of data presented.

We thank the reviewer for this advice. We have rewritten both the introduction and the discussion following this suggestion.

Reviewer #2:

The manuscript by Ishii et al reported that overactivation of Hsf1-Hsp signaling disturbed neuronal migration in the cerebral cortex which is in part a pathological mechanism underlying abnormalities in brain formation caused by developmental alcohol exposure. As the author stated, they previously reported reduction of Hsf1-Hsp signaling activity also induce developmental abnormalities. Over all, the authors successfully demonstrated discordant Hsf1-Hsp signaling play critical for the effect of environmental stress on brain development. The manuscript is very well written, but there are several issues need to be addressed.

We appreciate critical comments from the reviewer. We have performed new experiments and extending our discussion as according to the reviewer's suggestions, significantly improving and developing our manuscript.

The title of the current manuscript is too vague. Since this is research article, not review article, need modify it more specific (e.g., what signaling pathway, what prenatal stress) based on their findings in this study. Abstract should be re-written with more specific and conservative tone. Although the author tested several environmental stresses in Figure 1, most of the data were obtained in EtOH model. It is still unclear that specific mechanisms the authors presented here is also involved in developmental phenotypes caused by other stressors. The authors should also specify EtOH as a stressor (i.e., EtOH-induced Hsf1 overactivation) in result sections.

We thank the reviewer for the advice. Following the reviewer's suggestions, we have changed the title to "Variations in brain defects by prenatal alcohol exposure result from cellular mosaicism in the activation of heat shock signaling" to represent our findings more specifically. We have rewritten the abstract to provide more specificity based upon our findings. We have shown stochastic activation of heat shock signaling by H₂O₂ and MeHg *in vitro* (Fig. 1), and have also observed stochastic activation of heat shock signaling *in vivo* using the same reporter mice by other stressors (MeHg, suramin, maternal seizure, sodium arsenite etc.) similar to that by EtOH¹. In addition, prenatal exposure to suramin causes neuronal migration defects in reporter positive cells¹ similar to that induced by EtOH (Fig.3). Therefore, although EtOH is used as a representative stressor in this study, it is highly likely that other stressors share, at least in part, specific mechanisms presented here. One possible convergent mediator may be oxidative stress, which is commonly induced by these stressors. These points are now in the Result and Discussion in the revised manuscript (page 9-10, line 178-187; page 16, line 331-340).

Figure 3b, please show the comparative images of cellular distribution of Tuj1 neurons in EtOH-treated and PBS-treated mice. Only image of EtOH-treated mice is available in the current manuscript.

We have added the image accordingly.

In Figure 3g''', the authors successfully demonstrated delayed migration associated with HSP activity. Please present representative images of each condition with bin information. The distribution of RFP-positive cells looks very heterogeneous. How did the authors determine region of interest for quantitative analysis? There is no information of bin analysis for neuronal migration with ROI information in the method section which should be corrected.

We have added the required information in both the legend and the representative images in Figure 3.

Figure 5c

Distribution of GFP-positive cells in control treated with EtOH does not seem to be different compared with those in control without EtOH. Does EtOH have no effect on neuronal migration?

We previously reported that our regimen of 2.0 g/kg weight EtOH administration at E14-16 does not cause migratory deficits in most of the regions in the cerebral cortex². Most studies reporting migration deficits in the entire cortex have used heavier intoxication models, including our own work published in 2014³, in which chronic intoxication at E7.5-18.5 causes broad migration defects. We have added this note in the Methods section (under Drug Administration).

Figure 6 b', b'', d', d''

In the graph presented in Fig 6e, approximately 40% and 20% of GFP-positive cells are Cutl1-positive and BrdU-positive respectively. However, a very few subpopulation of GFP-positive cells looks Cutl1- and BrdU-positive, which is not consistent with the quantitative results. Please replace them to more representative images.

We apologize that these panels did not show the result clearly. GFP expression is found not only in cell bodies but also in neuronal processes that span in broader areas, often providing the false impression of a greater population GFP-positive cells than are present. We now provide higher magnification views to clearly visualize individual GFP⁺ positive cells, and indicate representative double-labeled cells with arrowheads.

In Figure 8, the author elegantly demonstrated importance of Hsf1 signaling pathway in neuronal migration defects caused by EtOH treatment. This is the key data supporting the authors' mechanistic hypothesis. One major concern is that the author did not demonstrate if overexpression of Hsf1 normalizes the effect of Hsf1 shRNA. Given that off-target effects are always concerns in shRNA experiments, the authors should test it to confirm that activation of Hsf1 is a mechanism of EtOH-induced migration defect.

We have added new data in Supplementary Fig. 9 which confirms on-target effects of *Hsf1* shRNA. In Hashimoto-Torii et al., 2014 *Neuron*², we similarly confirmed the on-target effects of the same *Hsf1* shRNA by an introduction of human *Hsf1* full-length *in vivo*.

Minor points;

Page 8 line 169: Did the author use DsRed2 protein? If not, please correct it.

We have changed the description to RFP (DsRed2) accordingly (line 136, line 173).

1. Torii M, *et al.* Detection of vulnerable neurons damaged by environmental insults in utero. *Proc Natl Acad Sci U S A*, (in press).
2. Hashimoto-Torii K, *et al.* Roles of heat shock factor 1 in neuronal response to fetal environmental risks and its relevance to brain disorders. *Neuron* **82**, 560-572 (2014).
3. El Fatimy R, *et al.* Heat shock factor 2 is a stress-responsive mediator of neuronal migration defects in models of fetal alcohol syndrome. *EMBO Mol Med* **6**, 1043-1061 (2014).

REVIEWERS' COMMENTS:

Reviewer #1 (Remarks to the Author):

The authors now provide compelling responses to all major issues that were raised in the initial review of this manuscript. Furthermore, the authors reorganized some of the figures and added new data which greatly strengthen the main conclusions of the manuscript. In my opinion, this manuscript adds significantly to our understanding of the underlying cause of the heterogeneous nature of focal cortical malformation in response to prenatal environmental stressors. Overall this study is likely to be of great interest to the broader neuroscience community.

Reviewer #2 (Remarks to the Author):

The authors have sufficiently answered all concerns from this reviewer.

We are pleased to hear that both reviewers found that we addressed most of their comments and their enthusiasm for our work.